# Electrochemotherapy Plus IL-2+IL-12 Gene Electrotransfer in Spontaneous Inoperable Stage III–IV Canine Oral Malignant Melanoma

**DOI:** 10.3390/vaccines11061033

**Published:** 2023-05-27

**Authors:** Matías Tellado, Mariangela De Robertis, Daniela Montagna, Daniela Giovannini, Sergio Salgado, Sebastián Michinski, Emanuela Signori, Felipe Maglietti

**Affiliations:** 1VetOncologia, Veterinary Oncology Clinic, Buenos Aires 1408, Argentina; mtellado@vetoncologia.com; 2Department of Biosciences, Biotechnology and Environment, University of Bari ‘A. Moro’, 70125 Bari, Italy; mariangela.derobertis@uniba.it; 3Instituto de Medicina Experimental (IMEX-CONICET), Academia Nacional de Medicina, Buenos Aires 1425, Argentina; daniela.r.montagna@gmail.com; 4ENEA SSPT-TECS-TEB, Casaccia Research Center, Division of Health Protection Technology (TECS), Agenzia Nazionale per le Nuove Tecnologie, l’Energia e lo Sviluppo Economico Sostenibile (ENEA), 00123 Rome, Italy; daniela.giovannini@enea.it; 5Laboratory of Molecular Pathology and Experimental Oncology, Institute of Translational Pharmacology, CNR, Rome 0133, Italy; 6CREOVet, Veterinary Oncology Clinic, Lima 04, Peru; sergiosndelp@gmail.com; 7Facultad de Medicina Veterinaria y Zootecnia, Universidad Peruana Cayetano Heredia, Lima 31, Peru; 8Instituto de Física Interdsiciplinaria y Aplicada (INFINA), Facultad de Cs Exactas y Naturales, UBA-CONICET, Buenos Aires 1428, Argentina; sebastianmichi@gmail.com; 9Instituto Universitario de Ciencias de la Salud, Fundación Barceló—CONICET, Buenos Aires 1117, Argentina

**Keywords:** electroporation, electrogene transfer, immune response, cancer, dog, ECT, EGT, immunotherapy

## Abstract

Electrochemotherapy (ECT) is a standard of care in veterinary and human oncology. The treatment induces a well-characterized local immune response which is not able to induce a systemic response. In this retrospective cohort study, we evaluated the addition of gene electrotransfer (GET) of canine IL-2 peritumorally and IL-12 intramuscularly to enhance the immune response. Thirty canine patients with inoperable oral malignant melanoma were included. Ten patients received ECT+GET as the treatment group, while twenty patients received ECT as the control group. Intravenous bleomycin for the ECT was used in both groups. All patients had compromised lymph nodes which were surgically removed. Plasma levels of interleukins, local response rate, overall survival, and progression-free survival were evaluated. The results show that IL-2 and IL-12 expression peaked around days 7–14 after transfection. Both groups showed similar local response rates and overall survival times. However, progression-free survival resulted significantly better in the ECT+GET group, which is a better indicator than overall survival, as it is not influenced by the criterion used for performing euthanasia. We can conclude that the combination of ECT+GET using IL-2 and IL-12 improves treatment outcomes by slowing down tumoral progression in stage III–IV inoperable canine oral malignant melanoma.

## 1. Introduction

Canine melanoma is the most frequent oral neoplasia diagnosed [1], and particularly in this location, melanoma is more aggressive and lethal [2]. Without treatment, it has a median survival of 65 days [3]. Dogs with pigmented oral mucosa, such as Cocker Spaniels and Poodles, are at an increased risk of developing this disease [4]. Local control of the disease is very important to ensure a good quality of life and reduce the incidence of metastasis. Radical surgery is the gold standard treatment, allowing very high response rates when clean resection margins are obtained [5], indicating that local therapy is of utmost importance for the prognosis of this disease. Chemotherapy does not increase local responses nor increase the survival of the patients [6]. In cases where radical surgery is not possible or is rejected by the owner, electrochemotherapy (ECT) or radiation therapy may be used [7]. As radiotherapy is costly and most of the time is not available in the veterinary setting, ECT emerges as a very valuable tool for inoperable cases, which can also be benefited from the addition of immunotherapies.

ECT is a locally enhanced chemotherapy used both in human and veterinary medicine. This treatment combines cell membrane electroporation and the administration of poorly permeating cytotoxic drugs, thereby improving drug delivery into the cells [8]. Bleomycin (BLM) and cisplatin are the only drugs with demonstrated efficacy for the procedure, both in human and veterinary medicine. BLM can be administered either systemically or intratumorally, and cisplatin only intratumorally. Carboplatin and oxaliplatin have been tested in combination with electroporation in cell lines with good results [9]. Many other chemotherapeutic drugs have been tested in vitro with variable results [10,11]. The electric parameters are eight square-wave monopolar 100 µs long pulses of 1000 V/cm at a repetition frequency of 1 to 5000 Hz [12,13,14]. In veterinary oncology, ECT is a very effective technique with no major side effects. Its main indications are cutaneous, subcutaneous, and mucosal tumors [15]. It is a highly effective treatment regardless of the histology of the tumors, thanks to the electroporation phenomenon. Being a physical method, electroporation induces cell membrane permeabilization allowing BLM to enter any cell type [16]. The commonly treated histologies are squamous cell carcinomas, malignant melanomas, sarcoids, soft tissue sarcomas, and mast cell tumors, where the treatment provides very high local response rates [17,18,19,20,21].

BLM’s mechanism of action depends on its intracellular concentration; it causes oxidative damage to the DNA, producing single-strand and double-strand breaks between 3′-4′ bonds in deoxyribose [22]. If few molecules are internalized, they induce G2-M arrest (slow mitotic cell death); thus, tumor tissues with elevated cell turnover are much more susceptible than normal ones. Otherwise, the entry of millions of BLM molecules into cells causes apoptosis [23]. It can induce immunogenic apoptosis in certain tumors [24], reactivate the antitumor immunity mediated by cytotoxic T cells [25], and reduces the expression of TGFβ or Tregs, both immunosuppressive [26]. Based on the studies mentioned above, BLM is recommended for ECT whenever possible [13,14].

In general, the antitumor effect of ECT is determined by three biological mechanisms. First, direct cytotoxicity is obtained by increasing the quantity of drug delivered to tumor cells [27]. Second, a transient vasoconstriction termed “vascular-lock”, instantly affects tumor blood flow, increasing tumor exposure to chemotherapy. It also has a delayed vascular disrupting action on tumor vessels, which results in tumor starvation, also contributing to cancer cell death [28]. Third, an immune stimulation, which is due to the release of damage-associated signals, triggers a strong stimulation of cancer immunity allowing to overcome tumor immune evasion [16,24]. This immune response improves treatment outcomes [16], and it is mainly of the humoral type, lacking the strength to affect distant metastasis. However, systemic properties of ECT have been recently described. Ruzgys et al. demonstrated an effect in non-treated tumors after ECT in mice [29]. Immunogenic tumor cell death can lead to the release of tumor antigens and further education of immune cells to attack distant tumor cells [30]. By switching the humoral response to a cellular response, it may be possible to increase the occurrence of the abscopal effect.

The systemic effect could be enhanced by immunotherapy using interleukins, such as interleukin-2 (IL-2) and interleukin-12 (IL-12). IL-2 was approved in the 1990s for the treatment of human metastatic melanoma and renal cell carcinoma and is still in use today in the recombinant protein form, administered intravenously or subcutaneously [31,32]. IL-2 is essential for the survival, proliferation, and differentiation of T lymphocytes and is considered a potent mitogen for them. However, IL-2 monotherapy shows limited effect in increasing survival rate due to high-dose-related toxicity and dual functional properties on T cells (activation of Treg cells) [33,34]. IL-12 is a proinflammatory cytokine responsible for the differentiation of Th1cells and induction of IFN-γ production, the acquisition of cytotoxic functions by CD8+ T cells, and enhancement of NK cell-mediated cytotoxicity [35]. The use of IL-2 and IL-12 for the treatment of melanoma has been extensively studied in preclinical models, including the delivery of plasmids encoding IL-2 and IL-12 using gene electrotransfer (GET) [36]. In the work of Komel et al., they showed that treating B16.F10 murine melanoma with GET using IL-2 and IL-12 plasmids administered intratumorally resulted in 71% complete responses [37]. Similar results were shown by Lucas et al.; they used the same mouse model and used GET with intratumoral IL-12. They obtained 47% of cures, and 70% of those cured mice showed resistance to the challenge with the injection of the same tumoral cells. In the same work, nude mice did not show any response. Interestingly, the intramuscular administration also did not show a response [38]. Significant tumoral growth delay was reported by Lohr et al. when transfecting B16.F10 tumors in mice using GET either with IL-2 or IL-12 [39].

Adjuvant immunotherapy has been used in feline vaccine-associated sarcoma [40]. The Oncept-IL-2 (Merial) includes an IL-2 plasmid that is vectored by a Canary Pox virus administered subcutaneously in the tumor site. It is used, for the treatment of feline fibrosarcomas, proving good efficacy and safety [41]. Regarding adjuvant treatment with Oncept (Merial) for canine oral malignant melanoma, the results are still a matter of debate [42,43].

GET using plasmids coding IL-2 in combination with IL-12 has been recently explored in veterinary medicine; they were administered either intratumorally, peritumorally, or intramuscularly [26]. This strategy could amplify the antitumor effectiveness of both cytokines [44]. They activate separate signaling pathways inducing complementary biological effects and lead to the expression of IFN-γ principally by T and NK cells and to the stimulation of IL-12 production by dendritic cells and macrophages [45]. GET using intratumoral delivery of IL-12 has achieved promising results in preclinical and clinical studies [45,46,47,48]. Dogs with advanced melanoma treated with this approach showed regression of melanoma lesions due to the activation of systemic cellular and humoral responses [49].

The stage is a very important prognostic factor for ECT as in early stages I and II, the objective response rates can be as high as 100% and 89%, respectively, while in late stages III and IV, response drops dramatically to 57% and 36%, respectively [17]. Combining ECT with GET in late-stage patients may increase the local immune response elicited by ECT and may induce an abscopal effect.

In this retrospective work, we compared the antitumor efficacy of ECT alone with ECT in combination with GET using plasmids coding canine IL-2 and IL-12 in dogs with spontaneous inoperable stage III–IV canine oral malignant melanoma.

## 2. Materials and Methods

### 2.1. Canine Patients

The medical records of three veterinary clinics that use GET, following the same protocol, were analyzed. The information on each patient included complete staging, treatments administered, and detailed follow-up regarding side effects, local response, progression-free survival (PFS), and overall survival (OS). As inclusion criteria, the patients enrolled had an oral malignant melanoma in stages III or IV and received ECT treatment. The ones without distant disease were excluded. Patients that only received ECT were included in the control group, while patients who received ECT+GET were included in the treatment group. Informed consent to include images and data of the patients in scientific research was signed by the respective owners. In Table 1, the subject’s demographics are shown.

The patients were staged according to the WHO staging system for canine oral malignant melanoma (see Table 2) [5].

Lymph node involvement was clinically assessed. When the inspection revealed signs of compromise, the lymph node was surgically removed and its compromise was determined by histopathology. Surgery was only performed to remove compromised lymph nodes. It is worth noting that all patients included in this work were not candidates for surgery with clean margins as it was not feasible.

### 2.2. Anesthetic Procedure

Chosen anesthesia protocol was proved to provide adequate comfort throughout the procedure and consisted of premedication with IM (intramuscular) administration of xylazine (Xylazine 100^®^, Richmond, Buenos Aires, Argentina). Induction was performed with IV (intravenous) administration of propofol (Propofol Gemepe^®^, Gemepe, Buenos Aires, Argentina) 2–3 mg/kg. For maintenance, isoflurane (Zuflax^®^, Richmond, Buenos Aires, Argentina) 2–3% and intravenous fentanyl (Fentanyl Gemepe^®^, Gemepe, Buenos Aires, Argentina) 2 mcg/kg were used. Amoxicillin with clavulanic acid (Clavamox^®^ Zoetis, Buenos Aires, Argentina) 15 mg/kg/bid and meloxicam (Meloxivet^®^, John Martin, Buenos Aires, Argentina) 0.2 mg/kg/SID were administered orally for prophylaxis and analgesia after the treatment according to the needs of each patient.

### 2.3. ECT Procedure

The Veterinary Guidelines for Electrochemotherapy of Superficial Tumors were followed. The patients were treated using intravenous BLM at a dose of 15,000 UI/m^2^ of BSA in bolus. Eight minutes after the administration of the drug, electric pulses were delivered using an EPV-200 electroporator (BIOTEX SRL, Buenos Aires, Argentina). The device delivers eight monopolar square-wave pulses, 100 μs long of 1000 V/cm at 5000 Hz on each train. These pulse parameters are effective regardless of the type of tissue treated and are the ones recommended by guidelines for human and veterinary patients [12,13,14,50]. The electrode used is a six-needle model composed of two rows of three needles separated 4 mm from each other. The needles were disposable.

### 2.4. GET Procedure

The GET procedure consisted of two transfections: one was performed the same day of the ECT, and the second one 28 days later. Two hundred micrograms of plasmid encoding canine IL-2 were injected into the periphery of the tumor, and then the GET pulses were delivered. After that, 200 μg of plasmid encoding canine IL-12 were injected into the quadriceps muscle, and then the GET pulses were delivered. To ensure that the plasmids were injected exactly where the pulses were applied, they were injected in between the needles of the electrode. For that purpose, the needle electrode was inserted halfway, then the needle of the syringe containing the plasmid was inserted between the needles of the electrode, and the plasmids were injected; then, the needle of the syringe was removed, and the electrode was completely inserted. Immediately after the complete insertion of the electrode, the electric pulses were delivered using the same electroporator. GET electric pulses consisted of eight monopolar square-wave 100 μs long pulses of 1000 V/cm at 1 Hz [51].

In Figure 1, a graphical description of the whole procedure is presented.

### 2.5. Plasmids

The pEGFP-N1 vector containing *Canis familiaris* IL-2 (Sequence ID: AM238655), IL-12A (Sequence ID: U49085.1), and IL-12B (Sequence ID: NM_001003292.1) genes were purchased from Cyagen Biosciences Inc. Kanamycin was used as bacterial resistance marker. The original vector was digested with XhoI (Promega Corporation, 2800 Woods Hollow Road Madison, WI, USA) to obtain a new plasmid containing only IL-12A/IL-12B genes. The IL-2 gene was amplified by PCR (Pfu polymerase, NEB) using the pEGFP-N1-IL2-IL12A-IL12B vector as a template. The following primers were used: Fw 5′-ATGTCGTAACAACTCCGCCC-3′ and Rev 5′-TGGTGTCTAGAAGAGGCCCG-3′. In the reverse primer, the restriction sequence for XbaI was introduced. The IL-2 PCR product was digested with XhoI and XbaI enzymes (Promega) and cloned in the pEGFP-N1 vector previously cut with the same enzymes. A new plasmid containing only the IL-2 gene was obtained. Each cloning was verified by sequencing. Isolation of both plasmid DNAs was performed using the Qiagen Endo-Free Kit (Qiagen^TM^, Hilden, Germany), according to the manufacturer’s instructions. Both pDNAs were diluted in endotoxin-free water (Qiagen^TM^) and checked by restriction analysis.

### 2.6. Blood Sample Collection and ELISA Test

Blood samples were extracted before the treatment, and every seven days after the transfection, the samples were centrifuged, and the plasma was stored at −18 °C. IL-2 and IL-12 concentration was measured using two canine-specific ELISA kits, Cusabio Biotech Co, Ltd. Dog Interleukin 2 (IL-2) ELISA KIT (Catalog number: CSB-E11258c, Cusabio, Houston, TX, USA) and MyBioSource Canine Interleukin 12 (IL-12) ELISA Kit (Catalog number: MBS742166, MyBioSource, Inc., San Diego, CA, USA), according to the manufacturer’s instructions. The absorbance at 450 nm was measured spectrophotometrically in a microplate reader. The ELISA determinations were performed in three replicates.

### 2.7. Evaluation of Treatment Outcome

The local response to treatment was assessed using the RECIST 1.1 criteria for solid tumors [52].

Considering the measurement of the longest tumor diameter, the response to treatment was classified in the following way. Complete Response (CR): the disappearance of all target lesions; Partial Response (PR): at least a 30% decrease in the sum of diameters of target lesions; Stable Disease (SD): a decrease of less than 30% or an increase of less than 20%; and Progressive Disease (PD): at least a 20% increase in the sum of diameters of target lesions, or the appearance of new lesions. Objective Responses (OR) are the sum of Complete and Partial responses. The response was evaluated one month after the treatment and confirmed one month afterward. Information was gathered at each follow-up visit, which included a chest X-ray, cervical ultrasound, observation, clinical palpation of lymph nodes, and an evaluation of the tumor, which was performed by the veterinary oncologist.

Side effects evaluated were vomiting, anorexia, pain, bleeding, inflammation, and respiratory distress, and were scaled according to the Veterinary Cooperative Oncology Group toxicity scale (VCOG-CTCAE) [53] in each follow-up visit.

OS and PFS were calculated from the moment that the ECT or ECT plus GET treatments were received until death or progression occurred, respectively.

### 2.8. Statistical Analysis

The calculation of the sample size and statistical analysis was performed using Medcalc 14.8.1. Fisher’s exact test was used to compare sex, objective response rate, and stage between groups. Mann–Whitney U was used to compare age and body weight between groups. OS and PFS between groups were compared using Kaplan–Meier curves, and the significance was assessed by the Log-rank test. A value of *p* < 0.05 was considered statistically significant.

## 3. Results

### 3.1. Treatment Groups Conformation

In total, 30 patients were included. Ten were in the ECT+GET treatment group, and twenty were in the ECT control group. There were no statistically significant differences among groups considering age (*p* = 0.453), body weight (*p* = 0.726), sex (*p* = 1.00), or stage (*p* = 0.419). Tumoral sizes were between 2–5 cm in both groups. In the ECT+GET group, 80% of the patients were in stage III and 20% in stage IV; in the ECT alone, 60% were in stage III and 40% in stage IV. All patients in both groups had lymph node involvement.

#### 3.1.1. Local Response

Considering the local response, patients undergoing ECT+GET treatment obtained an OR rate of 80% (CR rate of 30% and a PR rate of 50%), while patients in the control group obtained an OR rate of 65% (CR rate of 5% and a PR rate of 60%), although the differences in the response did not reach statistical significance (*p* = 0.674) (see Figure 2). In the ECT+GET group, the SD rate was 20% and the PD rate 0%, while in the control group, the SD rate was 25% and the PD rate 10%.

A case where a Complete Response in the ECT+GET group was obtained is shown in Figure 3.

#### 3.1.2. Overall Survival

The median survival of the patients in the ECT+GET group was 5.5 months (mean 10.3 months, range 3–32 months), while in the ECT group was 6 months (mean 6.45 months, range 2–17 months). The Kaplan–Meier curves revealed that these differences were not significant by the Log-rank test, *p* = 0.388 (Figure 4).

#### 3.1.3. Progression-Free Survival

The median PFS for the ECT+GET group was 5.5 months (mean 10.3 months, range 3–32 months), while for the ECT alone group was 4 months (mean 4.8 months, range 1–16 months). We observed that patients that added GET to the ECT treatment had statistically significantly better PFS (*p* = 0.0284) (Figure 5). The patients with the longest PFS also did not show progression at the end of the follow-up time and had the best survival times. 

### 3.2. Tolerance and Side-Effects

We observed muscle contractions in both groups during the application of the pulses, but the pain elicited was effectively controlled by the anesthesia. As expected in the ECT+GET group, the number of muscle contractions observed during pulse delivery was greater, as the pulses during GET protocol were applied at 1 Hz (below the response frequency threshold of the neuromuscular plaque [53,54]), producing eight contractions. Nevertheless, the GET procedure did not add any further discomfort or side effects.

After the treatment, side effects occurred within the first 5 days of the ECT or ECT+GET procedure. The most notable was pain, which was observed in 7/20 patients in the ECT group and in 3/10 patients in the GET group. In all cases, it was grade 1 or 2 (see Table 3). Grade 1 pain corresponds to mild pain not interfering with function, and grade 2 to moderate pain, moderately interfering with function or activities of daily life, where analgesic therapy is indicated. Vomiting was observed in both groups after the procedure and resolved by 8–12 h of fasting and by the administration of maropitant citrate 1mg/kg once a day (Cerenia, Zoetis, Buenos Aires, Argentina). Grade 1 vomiting corresponds to vomiting lasting less than 24 h that resolves with or without the use of medication and/or parenteral fluids.

Once the period of inflammation after the procedure resolved, no side effects were reported in any patient.

### 3.3. Expression of the Transfected Plasmids

Gene expression analysis revealed that the plasma concentration of IL-2 peaked on day 7, while IL-12 peaked on day 14 after transfection (Figure 6).

## 4. Discussion

In the treatment of canine oral malignant melanoma, the main objective is to obtain effective local control of the disease. This is commonly achieved with surgery with clean margins, and that often includes the bone margin. Lymphadenectomy of compromised lymph nodes is recommended [7]. ECT alone is a very good treatment option for the initial stages, when the tumor can be completely treated. However, in advanced stages, i.e., III and IV, the complete treatment of the lesion and margins with ECT is challenging [17]. In this scenario, GET is proposed as a treatment that can extend the effectiveness of ECT not only from a local point of view but from a systemic one. Theoretically, it can enhance the local immune response and reduce the occurrence of metastases that cannot be treated, as they are not accessible or not sensitive to other therapies. With this in mind, we considered that canine patients with oral malignant melanoma in stages III and IV are good candidates to test the hypothesis of combining ECT and GET since these cases are considered to have a systemic spread of the disease.

In these patients, it must be remembered that many factors may compromise the response to the ECT+GET treatment, so much as to make it difficult to obtain striking results. For example, there could be an evasion of the immune response at the primary site and at the site of metastases, the presence of infiltrate of immunosuppressive T lymphocytes, and we could apply the GET in a stage in which the immune evasion is irreversible.

Results in preclinical murine models are very promising combining ECT with IL-2 [55,56], ECT+IL-12 [57], and ECT with IL-2+IL-12 [37]; for that reason, the natural path is to move to a more realistic model, that is the dog with spontaneous tumors. This approach was already tested in dogs using GET but transfecting only IL-12 encoding plasmids [45,58]. Additionally, in human medicine, the use of GET with IL-12 rendered very promising results, achieving complete responses in 10% of the cases in a phase 1 study [46] and 11% of complete responses in a phase II study [59]. However, there are no studies regarding the combination of ECT and GET in human medicine yet. Moreover, the combination of ECT and GET using both cytokines has not been evaluated up to now, and as we can see in the results of this work, it has the potential to improve the treatment outcomes of ECT. A similar approach has been reported using ECT combined with immunotherapy using immune checkpoint inhibitors in the work of Campana et al. [60]. The authors evaluated the effect of adding ECT to Pembrolizumab treatment for cutaneous melanoma. They found that combining both treatments improved local response rates, OS, and PFS. As immune checkpoint inhibitors are not widely available and very expensive for veterinary medicine, GET could be an alternative to produce similar results. An advantage of GET is that it is easy to perform, virtually free of side effects, and relatively low cost, making immunotherapy more accessible to veterinary patients.

In this work, IL-2 was transfected peritumorally as the site of action for this cytokine is mainly local, and it can produce severe side effects when administered systemically if the dose is not correctly calculated [61]. Transfecting it to the peritumoral area increases the local availability of the protein and has similar effects to intratumoral transfection [62].

Regarding IL-12, this cytokine also showed toxicity when administered systemically; however, its expression was intended to be systemic to induce the switch of the immune response from humoral to cellular. For that reason, it was transfected to the muscle, as this transfection site tends to produce higher plasma concentrations of the protein [63,64].

Serum levels of IL-2 in healthy dogs range from 10.08 to 389.73 pg/mL [65], and IL-12 levels range from 3.26 to 29.46 pg/mL [66]. Maekawa et al. report serum levels of IL-2 ranging from 11.4 to 574 pg/mL and serum levels of IL-12 raging from 189.1 to 3026.7 pg/mL in dogs with malignant melanoma, with higher values of these cytokines correlating with better treatment outcomes [67]. The levels of these two cytokines are very variable among healthy or sick dogs; however, they play a central role in the prognosis and in the result of different treatments, as they reflect immunological responses in the tumor microenvironment [67]. The levels of IL-2 can be increased by the administration of BLM; however, the doses needed for producing that phenomenon are more than 10-fold the doses used in ECT [68]. Electroporation pulse parameters do not increase serum IL-2; however, when combined with BLM (ECT), it can induce slight but significant increases of it in the order of 14 pg/mL [29]. In the same work, the authors perform a GET with IL-2 encoding plasmid, observing the same pattern of expression that we report, with similar concentration increases in the order of 350 pg/mL. After 15 days, the serum concentration also drops significantly. In the patients treated with ECT+GET, we observed that the expression of both IL-2 and IL-12 plasmids started immediately after the transfection, and because of that, the concentration of the circulating proteins went from 0 to a peak between days 7 and 14. Consistent with reports from the literature [69,70,71,72], these data show a long-lasting release of IL-2 and IL-12 in serum, which could be associated with the reported effectiveness of the treatment. We observed a decrease in IL-2 levels after day 7 due to the low protein half-life in the serum [73]. Additionally, it is well characterized that helper T cell IL-2 production is limited by negative feedback also influenced by other common gamma chain family cytokines, i.e., IL-4, IL-7, IL-6, IL-12, and IL-27 [74]. We cannot rule out that the systemic expression of IL-12 may contribute to lowering the basal levels of IL-2 in serum by a negative feedback loop. A sustained high IL-12 concentration in serum could be attributed to functional cooperation between the two interleukins, which leads to a positive feedback loop over the IL-12 expression, as observed by Komel et al. [37].

On the one hand, local responses among the treatment and control groups have no statistically significant differences. It is well documented that the ECT provides excellent local disease control [75,76], thereby demonstrating additional benefits from the combination therapy requires the evaluation of larger case series. On the other hand, the extension in the PFS of the patients receiving ECT+GET can be considered as a slowing down of the disease progression [77,78]. Considering the results of other authors, in a study including nine canine patients with melanoma treated with ECT+GET with IL-12, the reported survival was 180 days [58]. This result is similar to the one obtained in our control group with ECT alone and also similar to our previous study [17].

Regarding the OS, we found very good survival times for both groups, and this is attributed to the good local control of the disease. Compared to other studies using ECT alone [17,58] or surgery alone [79], we found similar survival times. An important aspect is that PFS is a valid surrogate for OS [80], and in the case of veterinary medicine, this is very important. Contrary to human medicine, euthanasia is widely available and used when the condition of the patient is such that it is not tolerable. This situation may hinder the real OS times, as they are euthanized at time points where we cannot know for sure how long the patient will continue living. The OS, then, when euthanasia is possible, is determined by the decision of the owner and the veterinarian [81]. For this reason, we consider PFS a better indicator than OS for evaluating results in veterinary medicine.

Concerning the quality of life of the patients, we observed that in both treatment groups, the quality of life was better when an OR was obtained and when tumors were small [17]. This was in line with a previous study [82] showing the impact of different treatment modalities on it (ECT, GET, and ECT+GET vs. Surgery + ECT, Surgery + GET, and Surgery + ECT+GET).

Among the limitations of this work is the number of patients treated. It is possible that with a higher number of ECT+GET cases, statistical significance could have been achieved in the other aspects evaluated. Furthermore, the higher proportion of patients in stage IV in the control group can be considered a limitation, even though this difference was not statistically significant.

We consider that the next step to evaluate the benefit of ECT+GET is to treat stage IV patients, as the GET mainly aims to enhance the immune response against distant metastasis.

## 5. Conclusions

ECT is a very effective local therapy for treating canine oral malignant melanoma, which provides very good local responses in the advanced stages of the disease. The addition of GET with peritumoral canine IL-2 and intramuscular canine IL-12 can improve treatment outcomes by increasing the PFS without adding discomfort or side effects. This work encourages additional studies needed in this direction to elucidate the role of the combination of GET with ECT.

## Figures and Tables

**Figure 1 vaccines-11-01033-f001:**
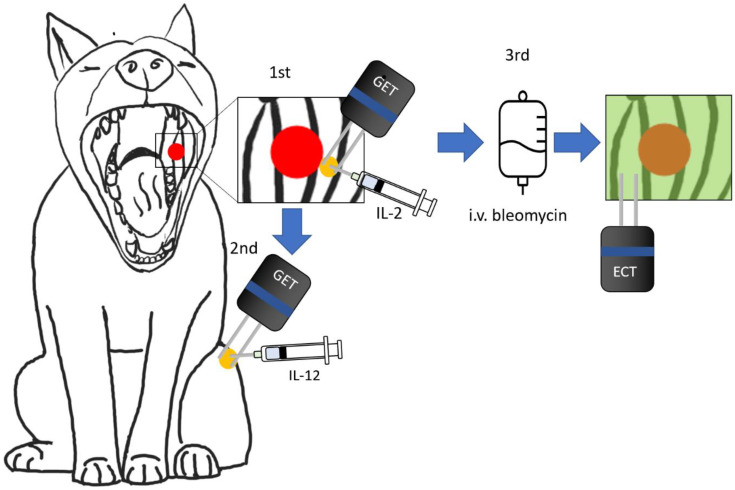
Schematic representation of the procedure in the ECT+GET treatment group. The tumor is represented with a red circle. The first step is to inject on the periphery of the tumor 200 μg of plasmid encoding canine IL-2, represented with a yellow circle, and then deliver the GET pulses. The second step is to inject 200 μg of plasmid encoding canine IL-12 in the quadriceps muscle (yellow circle) and then deliver the GET pulses. The third step is to treat with ECT using intravenous BLM, the tumor, and the safety margins (green area). Twenty-eight days later, steps 1 and 2 are repeated.

**Figure 2 vaccines-11-01033-f002:**
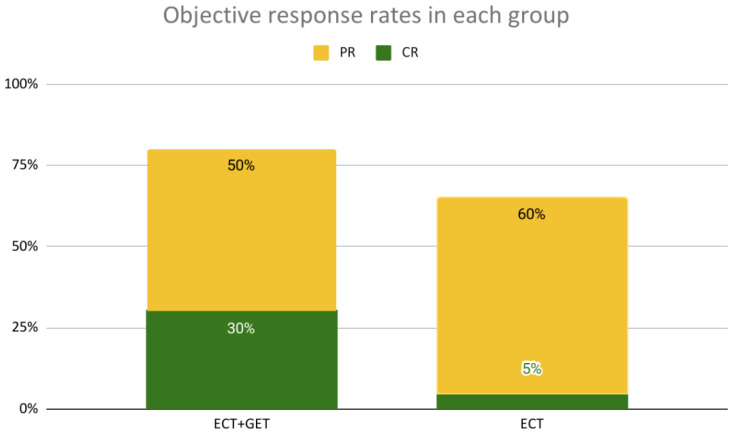
OR rates obtained in each group. Eighty percent of OR was obtained in the ECT+GET group and 65% in the ECT group (*p* = 0.674, Fisher’s exact test).

**Figure 3 vaccines-11-01033-f003:**
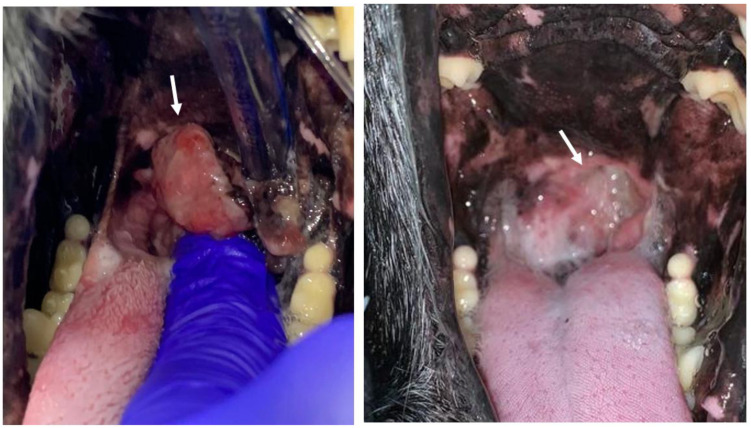
On the left, the patient with malignant melanoma located in the palate (white arrow) on the day of the ECT+GET treatment. On the right, 90 days after the treatment, a complete response was obtained. A fistula was formed due to the tumoral invasion of the whole depth of the palate. The patient, after a few days, adapted completely to the defect and had no symptoms.

**Figure 4 vaccines-11-01033-f004:**
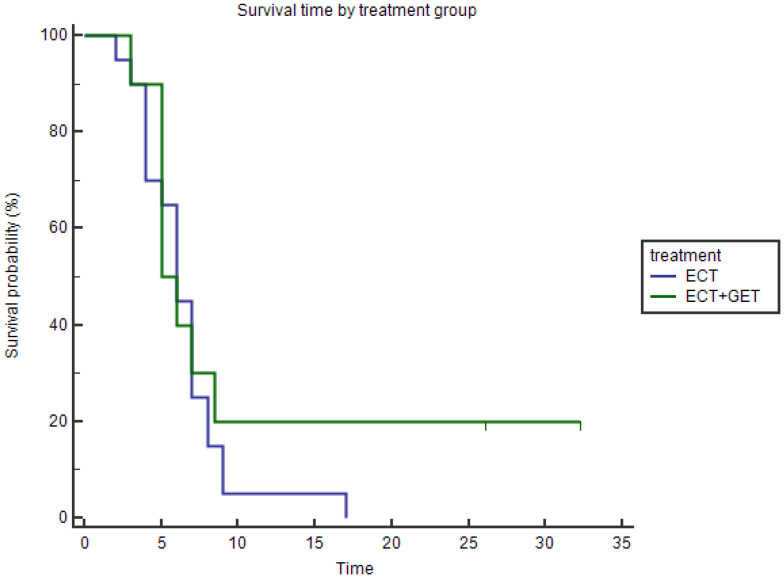
Kaplan–Meier curves for survival of the treatment groups. The blue line represents the ECT alone group, and the green line represents the ECT+GET group. The difference between groups was not statistically significant (*p* = 0.388, Log-rank test).

**Figure 5 vaccines-11-01033-f005:**
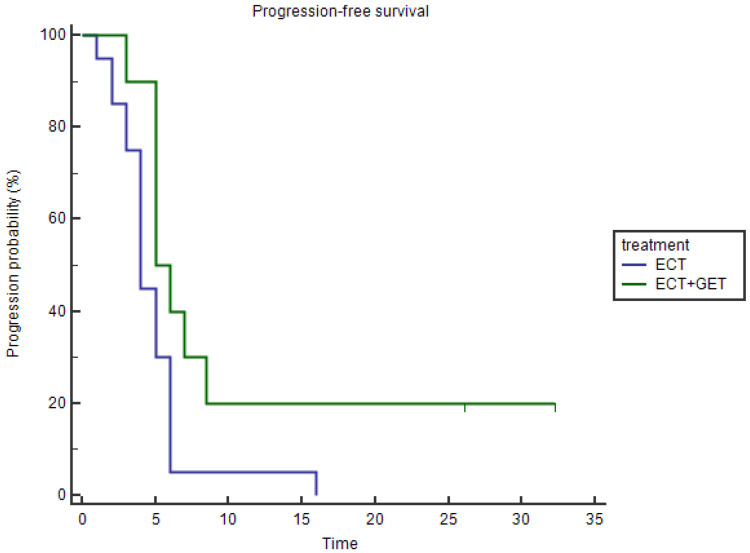
Kaplan–Meier curves for PFS are shown. The blue line represents the ECT alone group, and the green line represents the ECT+GET group. A longer PFS was achieved in the ECT+GET group (*p* = 0.0284, Log-rank test).

**Figure 6 vaccines-11-01033-f006:**
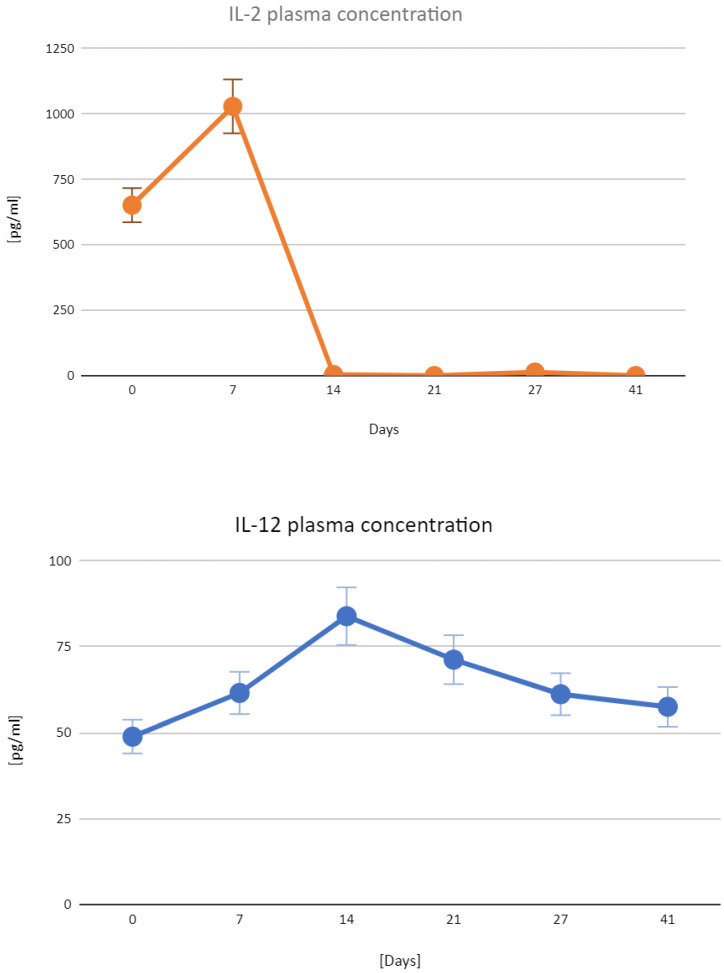
IL-2 and IL-12 concentration in plasma the days after the ECT+GET treatment.

**Table 1 vaccines-11-01033-t001:** Patient’s demographics.

Patients	ECT+GET	ECT
n (treatment: control)	10	20
Age (years)	11 (8–13)	11.8 (9–15)
Sex Ratio (F:M)	6:4	11:9
Body weight (kg)	23.8 (8–45)	23 (4–58)
Stage (III:IV)	8:2	12:8
Breeds		
Mixed breed	50%	30%
Beagle	10%	10%
Labrador Retriever	10%	10%
Golden Retriever	10%	0%
Shar-pei	10%	0%
Argentinian Dogo	10%	0%
Toy poodle	0%	20%
Rottweiler	0%	5%
Cocker Spaniel	0%	10%
English Mastiff	0%	5%
Dalmatian	0%	5%
Basset Hound	0%	5%

**Table 2 vaccines-11-01033-t002:** WHO staging system for canine oral malignant melanoma.

Stage	Tumor Diameter	Lymph Node Involvement	Metastasis
I	<2 cm	No	No
II	2–4 cm	No	No
III	> or =4 cm	No	No
	Any	Yes	No
IV	Any	Yes or No	Yes

**Table 3 vaccines-11-01033-t003:** Side-effects in both groups according to the VCOG-CTCAE v2. As can be seen, only grade 1/2 side effects were present.

	Grade 1		Grade 2	
Treatment group	ECT	ECT+GET	ECT	ECT+GET
Pain	5 (25%)	2 (20%)	2 (10%)	1 (10%)
Vomiting	2 (10%)	2 (20%)	-	-

## Data Availability

Data are available upon request.

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
