# Peer review of "Electrochemotherapy Plus IL-2+IL-12 Gene Electrotransfer in Spontaneous Inoperable Stage III–IV Canine Oral Malignant Melanoma"

_vaccines, 2023, doi:10.3390/vaccines11061033_

Round 1

Reviewer 1 Report

A brief summary 

In the paper, authors investigated the combinatorial treatment including electrochemotherapy and gene electrotransfer of two plasmids encoding canine IL-2 and IL-12. The clinical study compared the efficacy of the therapy in canine oral melanomas of high stages. The survival of the animals after ECT and ECT+IL-12+IL-2 was compared. However, the progression-free survival was longer after the combination therapy. Additionally, the serum concentration of the interleukins were evaluated, supporting the transfection efficiency and expression profile in time. Results are promising; however, not supported by the preclinical data.

General concept comments

The theme is interesting and the results are promising. However, there are numerous general and specific problems in this manuscript, from basic grammar and typing problems, flaws in experimental design and in the reporting of the results, to applicability of the findings. The results should be reported more systematically, and discussion extended to interpret the results more clearly. Although there is some valuable data to be published from this research, the manuscript, as it is, is weak and unrefined.

The clinical study should be supported by the preclinical data. Conclusions are poor. The ethical statement/clinical study permission is missing.

Specific comments

Abstract

-It is not specified how the IL-2 and IL-12 plasmids are delivered. Is the application intratumoral, intradermal, subcutaneous…? Please specify (throughout the paper).

-Specify the chemotherapeutic drug used in ECT.

-Line 32, 34: Which groups/treatments? Please specify.

-“Progression-free survival was significantly better in the treatment group, being a better indicator than overall survival for these cases as euthanasia can hinder the overall survival times.« Please rephrase.

Keywords

In the title, »gene electrotransfer« is used. Contrary, in the keywords, »electrogene transfer«. Please unify.

Background

There are many paragraphs, which is somehow confusing. I recommend to join some of them.

-Line 58: Please specify that depending on the tissue, which we are trying to permeabilize by electroporation, different electric pulse parameters are used. Add references of the SOP for ECT in clinics.  

-63: Please point out, which ECTs with which cytostatics and applications (i.t./i.v.) are used in veterinary oncology.

-73: Describe all the types of cell death by BLM or ECT/BLM

-91: Only BLM is described here. What else is available in veterinary oncology? The recommendation (that you stated) is not possible without evaluation of the other possibilities/cytostatics.

-97: The abscopal effect is rare; however, it was proved after ECT in preclinical studies in mice. Please add the references.

-When describing IL-2 and IL-12 therapies, always specify the type of the therapy (protein, gene therapy…) and the delivery method (electroporation, viruses, i.v., i.t….)

Materials and Methods

-Table 1: I do not understand the data on breeds for ECT group.

-How did you determine that the LNs are metastatic? Please specify.

-164: “The GET protocol used in this work was previously validated in vitro in 3T3-L1 cells, transfecting a plasmid encoding a green fluorescent protein (GFP). Thirty percent of cells expressed the fluorescent protein. This result was similar to the one obtained using lipofectamine (data not shown).” I would skip the text or describe the in vitro work more precisely and add the results. The translation from in vitro to in vivo should be more precisely supported by the data.

-»The GET procedure consisted of two transfections, one was performed the same day of the ECT, and the second one 28 days later.« Please describe the protocol in detail. Which plasmid was injected firs? I would also add a schematic representation of the treatment. Please specify the type of the muscle transfected and what does “periphery of the tumor« mean.

-ELISA: When were the samples taken on day 0, before the therapy or after the therapy?

-197: The three (technical) replicates?

Results

-236: Which differences, which groups? Please specify.

-246: Where is the Table 5? Is it anything missing in the caption?

-250: Since the results are not significant, you cannot state that the survival of the patients treated with ECT+GET was longer.

-269, 270: Which side effects did you evaluate? Please specify.

-Figure 4 caption: specify the treatment group.

-Serum concentration of IL-2 and IL-12: Why didn’t you measure the concentration of the IL-12 on day 0? What are the serum concentrations of IL-2 and IL-12 in the control group (ie healthy dog)?

-I miss the results about LN

Discussion

-310: Review the literature about the preclinical evaluation of ECT+IL-2+IL-12 that would support the further use in the clinical setting.

-320: Discuss how the application site (muscle, periphery of the tumor) affect the expression of the protein in serum. Why did you decide to perform intramuscular and tumor periphery GET?  Why didn’t you measure the concentration of the IL-12 on day 0? Why the concentration of IL-2 is higher on day 0 in comparison to day >14?

-321. 346: This is not scientific language. Support the results with the statistical analyses.

-Discuss about the limitations of the study.

Conclusion

Poor, not scientific.

Moderate editing of English is required

Author Response

Responses to the reviewer in bold

A brief summary 

In the paper, authors investigated the combinatorial treatment including electrochemotherapy and gene electrotransfer of two plasmids encoding canine IL-2 and IL-12. The clinical study compared the efficacy of the therapy in canine oral melanomas of high stages. The survival of the animals after ECT and ECT+IL-12+IL-2 was compared. However, the progression-free survival was longer after the combination therapy. Additionally, the serum concentration of the interleukins were evaluated, supporting the transfection efficiency and expression profile in time. Results are promising; however, not supported by the preclinical data.

We would like to thank the reviewer for taking the time to revise the manuscript and provide very valuable observations that allowed us to improve the manuscript.
Regarding preclinical data that support the present work, the following text was added to the introduction:

“The use of IL-2 and IL-12 for the treatment of melanoma has been extensively studied in preclinical models, including the delivery of plasmids encoding IL-2 and IL-12 using gene electrotransfer (GET) [36]. In the work of Komel et al., they showed that treating B16.F10 murine melanoma with GET using IL-2 and IL-12 plasmids administered intratumorally resulted in 71% complete responses[37]. Similar results were shown by Lucas et al., they used the same mouse model and used GET with intratumoral IL-12. They obtained 47% of cures, and 70% of those cured mice showed resistance to the challenge with the injection of the same tumoral cells. In the same work, nude mice did not show any response. Interestingly, the intramuscular administration also did not show a response [38]. Significant tumoral growth delay was reported by Lohr et al., when transfecting B16.F10 tumors in mice, using GET  either with IL-2 or IL-12[39].”

The present work can be considered either a veterinary clinical or a preclinical study for human medicine. We used species-specific ILs in spontaneous tumors, which show a higher possibility of translation from veterinary to human medicine. 

General concept comments

The theme is interesting and the results are promising. However, there are numerous general and specific problems in this manuscript, from basic grammar and typing problems, flaws in experimental design and in the reporting of the results, to applicability of the findings. The results should be reported more systematically, and discussion extended to interpret the results more clearly. Although there is some valuable data to be published from this research, the manuscript, as it is, is weak and unrefined.

The clinical study should be supported by the preclinical data. Conclusions are poor. The ethical statement/clinical study permission is missing.

We agree that the manuscript has valuable data that should be published, all the corrections had been made to refine and improve the manuscript. Also, the English were thoroughly revised. 

Informed consent was signed by the owners, as stated in the ethical statement section. The model of it was submitted later upon the editor's request. 

Specific comments

Abstract

-It is not specified how the IL-2 and IL-12 plasmids are delivered. Is the application intratumoral, intradermal, subcutaneous…? Please specify (throughout the paper).

The IL-2 was delivered intratumorally, and the IL-12 intramuscularly. These details were added to the abstract, and specified throughout the manuscript. For the sake of clarity we added a new figure  (now indicated as Figure 1) with a graphical description of the procedure, and the text in the materials and methods section was improved. 

-Specify the chemotherapeutic drug used in ECT.

Following the recommendation of the veterinary guidelines for electrochemotherapy we used i.v. Bleomycin, this information was added to the abstract. 

-Line 32, 34: Which groups/treatments? Please specify.

The abstract was rewritten describing the two groups, which were ECT+GET as the treatment group, and ECT alone as the control group. 

-“Progression-free survival was significantly better in the treatment group, being a better indicator than overall survival for these cases as euthanasia can hinder the overall survival times.« Please rephrase.

 The sentence was rephrased and now reads:

“However, progression-free survival resulted significantly better in the ECT+GET group, which is a better indicator than overall survival, as is not influenced by the criterion used for performing euthanasia.” 

Keywords

In the title, »gene electrotransfer« is used. Contrary, in the keywords, »electrogene transfer«. Please unify.

In the literature, the technique can be found with either of those two names. To increase the chances of the work being found in search engines, we followed recommendations from Gbur and Trumbo, of not repeating words from the title in the keywords section. 

[From Section 5, Suggestions for authors: selecting key words and phrases, pp. 31–32 from ‘Key words and phrases – the key to scholarly visibility and efficiency in an information explosion’, by E.E. Gbur and B. Trumbo, pp. 29–33, Vol. 49(1), June 1995. 

Hartley, James; Kostoff, Ronald N. (2003). How Useful are `Key Words' in Scientific Journals?. Journal of Information Science, 29(5), 433–438. doi:10.1177/01655515030295008]

Background

There are many paragraphs, which is somehow confusing. I recommend to join some of them.

The paragraphs were grouped according to the topics.

-Line 58: Please specify that depending on the tissue, which we are trying to permeabilize by electroporation, different electric pulse parameters are used. Add references of the SOP for ECT in clinics.  

Before the publication of the human SOP for ECT, many pulse parameters were used, depending on tumor type and tissue treated, obtaining variable results. However, after the SOP publication, the recommended pulse parameters were unified. The ones used in this work are effective regardless of the type of tissue treated and are the ones that are recommended by human SOP and veterinary guidelines. The mentioned publications are the following:

Gehl, Julie, et al. "Updated standard operating procedures for electrochemotherapy of cutaneous tumours and skin metastases." Acta oncologica 57.7 (2018): 874-882.

Mir, Lluis M., et al. "Standard operating procedures of the electrochemotherapy: Instructions for the use of bleomycin or cisplatin administered either systemically or locally and electric pulses delivered by the CliniporatorTM by means of invasive or non-invasive electrodes." European Journal of Cancer Supplements 4.11 (2006): 14-25.

Tozon, Natasa, et al. "Operating procedures of the electrochemotherapy for treatment of tumor in dogs and cats." JoVE (Journal of Visualized Experiments) 116 (2016): e54760.

Tellado, Matías, Lluis M. Mir, and Felipe Maglietti. "Veterinary Guidelines for Electrochemotherapy of Superficial Tumors." Frontiers in Veterinary Science 9 (2022).

We added a brief comment in line 58: “These pulse parameters are effective regardless of the type of tissue treated and are the ones recommended by guidelines for human and veterinary patients [12–14,50]” 

-63: Please point out, which ECTs with which cytostatics and applications (i.t./i.v.) are used in veterinary oncology.

Electrochemotherapy can be performed using only two cytostatics. Bleomycin administered i.t or iv., or cisplatin, administered i.t. The sentence regarding this was rephrased and now reads:

“Bleomycin (BLM) and cisplatin are the only drugs with demonstrated efficacy for the procedure, both in human and veterinary medicine. BLM can be administered either systemically or intratumorally, and cisplatin only intratumorally.”

-73: Describe all the types of cell death by BLM or ECT/BLM

The cell death types induced by BLM and ECT/BLM are essentially the same, as ECT enhances the entrance to the cell, but does not affect the mechanism of action. Additionally, ECT is a more potent stimulator of the immune system than electroporation or BLM alone, and in that sense, the immunological effects are increased. Extended cell-cycle arrest, apoptosis, and mitotic cell death are the most common outcomes of bleomycin treatment, which are described in the following paragraph:

“BLM’s mechanism of action depends on its intracellular concentration, it causes oxidative damage to the DNA producing single-strand and double-strand breaks between 3'-4' bonds in deoxyribose [22]. If few molecules are internalized, they induce G2-M arrest (slow mitotic cell death), thus tumor tissues with elevated cell turnover are much more susceptible than normal ones. Otherwise, the entry of millions of BLM molecules into cells causes apoptosis [23] “

-91: Only BLM is described here. What else is available in veterinary oncology? The recommendation (that you stated) is not possible without evaluation of the other possibilities/cytostatics.

As mentioned before, ECT can be performed with bleomycin or cisplatin. Cisplatin is reserved for very small tumors, and intravenous bleomycin is the drug and administration route recommended for most of the cases, and for that reason, only BLM is described. Recently calcium has been combined with electric pulses with very good results, however, the therapy is termed calcium electroporation, and for that reason is not included in the electrochemotherapy description. Regarding other drugs, a very extensive list of drugs has been tested in combination with electroporation, with minimum to no advantage of their combination.  

Regarding other treatment options available in veterinary medicine, it is quite similar to human medicine with cost limitations. In the particular case of melanoma, immune checkpoint inhibitors are not available, and for that reason, we present GET as a cost-effective alternative. 

-97: The abscopal effect is rare; however observe, it was proved after ECT in preclinical studies in mice. Please add the references.

The following text was added to the introduction:

“The local immune response elicited by ECT improves treatment outcomes [16], and it is mainly of the humoral type, lacking the strength to affect distant metastasis. However, systemic properties of ECT have been recently described. Ruzgys et al. demonstrated an effect in non-treated tumors after ECT in mice [29]”

-When describing IL-2 and IL-12 therapies, always specify the type of the therapy (protein, gene therapy…) and the delivery method (electroporation, viruses, i.v., i.t….)

The type of therapies and way of administration were added when mentioned in the manuscript. 

Materials and Methods

-Table 1: I do not understand the data on breeds for ECT group.

There was an error in the data of the Table, it was corrected.

-How did you determine that the LNs are metastatic? Please specify.

The following sentence was added to the material and methods section: 

“Lymph node involvement was clinically assessed, and when the inspection revealed signs of compromise, the nodule was surgically removed and determined its compromise by histopathology.  “

-164: “The GET protocol used in this work was previously validated in vitro in 3T3-L1 cells, transfecting a plasmid encoding a green fluorescent protein (GFP). Thirty percent of cells expressed the fluorescent protein. This result was similar to the one obtained using lipofectamine (data not shown).” I would skip the text or describe the in vitro work more precisely and add the results. The translation from in vitro to in vivo should be more precisely supported by the data.

We agree with the reviewer, the in vitro work deserves a publication for itself. For that reason, we decided to remove the paragraph from the manuscript. 

-»The GET procedure consisted of two transfections, one was performed the same day of the ECT, and the second one 28 days later.« Please describe the protocol in detail. Which plasmid was injected firs? I would also add a schematic representation of the treatment. Please specify the type of the muscle transfected and what does “periphery of the tumor« mean.

The procedure description was improved, and a new figure was added for the sake of clarity.

Figure 1. Schematic representation of the procedure in the ECT+GET treatment group. The tumor is represented with a red circle. The first step is to inject on the periphery of the tumor 200 μg of plasmid encoding canine IL-2 represented with a yellow circle, and then deliver the GET pulses. The second step is to inject 200μg of plasmid encoding canine IL-12 in the quadriceps muscle and then deliver the GET pulses. The third step is to treat the tumor and safety margins with electrochemotherapy using intravenous bleomycin. Twenty-eight days later, steps 1 and 2 are repeated.

-ELISA: When were the samples taken on day 0, before the therapy or after the therapy?

The samples were taken before the therapy.  We have clarified this point in the text 

-197: The three (technical) replicates?

 The sentence was corrected and now states:

“The ELISA determinations were performed in three replicates.” 

Results

-236: Which differences, which groups? Please specify.

The groups were specified correcting the following sentence:

“In total, 30 patients were included. Ten were in the ECT+GET treatment group and 20 were in the ECT control group.”

-246: Where is the Table 5? Is it anything missing in the caption?

We detected an error in the word file, it should not read “Table 5”, the correct sentence is: 

“The median survival of the patients in the ECT+GET group was 5.5 months (mean 10.3 months, range 3-32 months), while in the ECT group was 6 months (mean 6.45 months, range 2-17 months). “

It was corrected in the manuscript. 

-250: Since the results are not significant, you cannot state that the survival of the patients treated with ECT+GET was longer.

We agree with the reviewer. The sentence regarding the difference in the OS was removed.

-269, 270: Which side effects did you evaluate? Please specify.

In the materials and methods section, the evaluated side-effects were added:

“Side effects evaluated were vomiting, anorexia, pain, bleeding, inflammation, and respiratory distress, and scaled according to the Veterinary Cooperative Oncology Group toxicity scale (VCOG-CTCAE) [51] in each follow-up visit.”

-Figure 4 caption: specify the treatment group.

The caption of Figure 4 (now Figure 5) was corrected. Now it reads

“Figure 5. Kaplan-Meier curves for PFS are shown. The blue line represents the ECT alone group and the green line represents the ECT+GET group. A longer PFS was achieved in the ECT+GET group (p=0.0284, Log-rank test).”

-Serum concentration of IL-2 and IL-12: Why didn’t you measure the concentration of the IL-12 on day 0? What are the serum concentrations of IL-2 and IL-12 in the control group (ie healthy dog)?

As this is a retrospective study, and IL-2 and IL-12 concentrations are not routinely measured, those values from the control group are not available. However, we added the following information to the discussion:  

“Serum levels of IL-2 in healthy dogs range from 10.08  to 389.73 pg/ml [65], and IL-12 levels range from 3.26 to 29.46 pg/mL [66]. Maekawa et al. report serum levels of IL-2 ranging from 11,4 to 574 pg/ml, and serum levels of IL-12 raging from 189,1 to 3026,7 pg/ml in dogs with malignant melanoma, with higher values of these cytokines correlating with better treatment outcomes[67]. The levels of these two cytokines are very variable among healthy or sick dogs, however, they play a central role in the prognosis and in the result of different treatments, as they reflect immunological responses in the tumor microenvironment[67].]” 

The measurement of IL-12 on day 0 (prior to treatment) was missing from the graph, and it has been incorporated. 

-I miss the results about LN 

The results about LN compromise were in the materials and methods section, and were moved to the results section: 

“All patients in both groups had LN involvement.”

Discussion

-310: Review the literature about the preclinical evaluation of ECT+IL-2+IL-12 that would support the further use in the clinical setting. 

From the perspective of human medicine, we consider this work a preclinical trial. Considering the veterinary setting, both ILs have been used separately with good results, however by different means of administration. The following paragraph was extended to clarify this

“Results in preclinical murine models are very promising combining ECT with IL-2[55,56], ECT+IL-12 [57], and ECT with IL-2+IL-12 [37], for that reason the natural path is to move to a more realistic model, that is the dog with spontaneous tumors. This approach was already tested in dogs using GET, but transfecting only IL-12 encoding plasmids [45,58]. Also, in human medicine the use of GET with IL-12 rendered very promising results, achieving complete responses in 10% of the cases in a phase 1 study[46] and 11% of complete responses in a phase II study[59]. But there are no studies regarding the combination of ECT and GET in human medicine yet. Moreover, the combination of ECT and GET using both cytokines has not been evaluated up to now, and as we can see in the results of this work, it has the potential to improve the treatment outcomes of ECT. A similar approach has been reported using ECT combined with immunotherapy using immune checkpoint inhibitors in the work of Campana et al.[60]. The authors evaluated the effect of adding ECT to Pembrolizumab treatment for cutaneous melanoma. They found that combining both treatments improved local response rates, OS, and PFS. As immune checkpoint inhibitors are not widely available and very expensive for veterinary medicine, GET could be an alternative to produce similar results. An advantage of GET is that is easy to perform, virtually free of side effects, and relatively low cost, making immunotherapy more accessible to veterinary patients.”

-320: Discuss how the application site (muscle, periphery of the tumor) affect the expression of the protein in serum. Why did you decide to perform intramuscular and tumor periphery GET?  Why didn’t you measure the concentration of the IL-12 on day 0? Why the concentration of IL-2 is higher on day 0 in comparison to day >14?

We found the observations of the referee very interesting and decided to extend the following paragraph of the discussion

“In the patients treated with ECT+GET, we observed that the expression of both IL-2 and IL-12 plasmids started immediately after the transfection, and because of that, the concentration of the circulating proteins went from 0 to a peak between days 7 and 14. Consistent with reports from the literature [69],[70],[71],[72], these data show a long-lasting release of IL-2 and IL-12 in serum, which could be associated with the reported effectiveness of the treatment. In particular, we observed a decrease in IL-2 levels after day 7, due to the low protein half-life in the serum[73]. Additionally, it is well characterized that helper T cell IL-2 production is limited by negative feedback influenced also by other common gamma chain family cytokines, i.e. IL-4, IL-7, IL-6, IL-12, and IL-27[74]. We cannot rule out that the systemic expression of IL-12 may contributed to lower the basal levels of IL-2 in serum by a negative feedback loop. A sustained high IL-12 concentration in serum could be attributed to functional cooperation between the two interleukins, which leads to a positive feedback loop over the IL-12 expression, as observed by Komel et al.[37]“

The graphic of IL-12 had a missing point at day 0, which was added.

Regarding transfection sites the following paragraph was added.

“In this work, IL-2 was transfected peritumorally as the site of action for this cytokine is mainly local, and it can produce severe side effects when administered systemically if the dose is not correctly calculated [61]. Transfecting it to the peritumoral area increases the local availability of the protein, and has similar effects to the intratumoral transfection[62].

Regarding IL-12, this cytokine also showed toxicity when administered systemically, however, its expression was intended to be systemic to induce the switch of the immune response from humoral to cellular. For that reason, it was transfected to the muscle, as this transfection site tends to produce higher plasma concentrations of the protein[63,64].” 

-321. 346: This is not scientific language. Support the results with the statistical analyses.

We agree that in the mentioned sentences of the manuscript, we are speculating about the reasons why these results were obtained, and they were excessively optimistic. We rephrased them to reflect more concisely the results. The paragraph was rewritten and now states:

“On the one hand, local responses among the treatment and control groups have no statistically significant differences. It is well documented that the ECT provides excellent local disease control [75,76], therefore demonstrating additional benefits from the combination therapy requires the evaluation of larger case series. On the other hand, the extension in the PFS of the patients receiving  ECT+GET can be considered as a slowing down of the disease progression [77,78]. Considering the results of other authors, in a study including 9 canine patients with melanoma treated with ECT+GET with IL-12, the reported survival was 180 days[58]. This result is similar to the one obtained in our control group with ECT alone, and also similar to our previous study[17].

Regarding the OS, we found very good survival times for both groups and this is attributed to the good local control of the disease. Compared to other studies using ECT alone[17][58] or surgery alone[79] we found similar survival times. An important aspect is that PFS is a valid surrogate for OS[80], and in the particular case of veterinary medicine, this is very important. Contrary to human medicine, euthanasia is widely available and used when the condition of the patient is such that it is not tolerable. This situation may hinder the real OS times, as they are euthanized at time points where we cannot know for sure how long the patient will continue living. The OS then, when euthanasia is possible, is determined by the decision of the owner and the veterinarian[81]. For this reason, we consider PFS a better indicator than OS for evaluating results in veterinary medicine.” 

-Discuss about the limitations of the study.

 The limitations of the study were added. Now the end of the section reads:

“Among the limitations of this work is the number of patients treated. It is possible that with a higher number of ECT+GET cases, statistical significance could have been achieved in the other aspects evaluated. Furthermore, the higher proportion of patients in stage IV in the control group can be considered a limitation,  even though this difference was not statistically significant.”

Conclusion

Poor, not scientific.

We completely rewrote the conclusion, now it reads:

“ECT is a very effective local therapy for treating canine oral malignant melanoma, which provides very good local responses in advanced stages of the disease. The addition of GET with peritumoral canine IL-2 and intramuscular canine IL-12 can improve treatment outcomes by increasing the PFS without adding discomfort or side effects. This work encourages additional studies needed in this direction to elucidate the role of the combination of GET with ECT.”

Reviewer 2 Report

How were the patient selected? What were the inclusion and exclusion criteria? This should be clearly presented.

Was any sample size analysis performed?

The groups are not comparable: 20 % of the patients in the ECT+GET group were in stage IV compared to 40 % in the ECT only group.

The difference in progressive free survival between the groups could be a consequence of this difference in stage. The results does not support the conclusion.

Not all responses after treatment are reported. Did the rest of the patients have progressive disease?

What are normal variation in IL-2 and IL-12 after ECT treatment alone, do they differ from ECT+EGT?

A language review is recommended. 

Author Response

Response to reviewer in bold

Comments and Suggestions for Authors

How were the patient selected? What were the inclusion and exclusion criteria? This should be clearly presented.

We would like to thank the reviewer for taking the time to revise the manuscript and make valuable comments to improve it.

The following text was added to the materials and methods section:

“The medical records of three veterinary clinics that use GET, following the same protocol, were analyzed. The information on each patient included complete staging, treatments administered, and detailed follow-up regarding side effects, local response, PFS, and survival. As inclusion criteria, the patients enrolled had an oral malignant melanoma in stages III or IV and received ECT treatment. The ones without distant disease were excluded.”  

Was any sample size analysis performed?

In this retrospective study, we selected the last 10 patients treated with GET which comply with the inclusion criteria. And for the control group, the patients in the same period of time and medical situation that had a complete medical record and follow-up. For estimating sample sizes, the MedCalc software was used. This information was added to the manuscript. 

The groups are not comparable: 20 % of the patients in the ECT+GET group were in stage IV compared to 40 % in the ECT only group.The difference in progressive free survival between the groups could be a consequence of this difference in stage. The results does not support the conclusion.

To address this concern of the referee we confirm that the number of patients in stage IV in both groups have no statistically significant differences (Fisher test, p=1,00). 

Not all responses after treatment are reported. Did the rest of the patients have progressive disease?

The responses of all the patients were included in the manuscript. The following sentence was added: 
“In the ECT+GET group, the SD rate was 20% and the PD rate 0%, while in the control group, the SD rate was 25% and the PD rate 10%. “

What are normal variation in IL-2 and IL-12 after ECT treatment alone, do they differ from ECT+EGT?

The following paragraph of the discussion has been extended to address this question:

“Serum levels of IL-2 in healthy dogs range from 10.08  to 389.73 pg/ml [65], and IL-12 levels range from 3.26 to 29.46 pg/mL [66]. Maekawa et al. report serum levels of IL-2 ranging from 11,4 to 574 pg/ml, and serum levels of IL-12 raging from 189,1 to 3026,7 pg/ml in dogs with malignant melanoma, with higher values of these cytokines correlating with better treatment outcomes[67]. The levels of these two cytokines are very variable among healthy or sick dogs, however, they play a central role in the prognosis and in the result of different treatments, as they reflect immunological responses in the tumor microenvironment[67]. The levels of IL-2 can be increased by the administration of BLM, however, the doses needed for producing that phenomenon are more than 10-fold the doses used in ECT[68]. Electroporation pulse parameters do not induce the increase of serum IL-2, however, when combined with BLM (ECT), it can induce slight but significant increases of it in the order of 14 pg/ml[29]. In the same work, the authors perform a GET with IL-2 encoding plasmid, observing the same pattern of expression that we report, with similar concentration increases in the order of 350 pg/ml. After 15 days, the serum concentration also drops significantly. In the patients treated with ECT+GET, we observed that the expression of both IL-2 and IL-12 plasmids started immediately after the transfection, and because of that, the concentration of the circulating proteins went from 0 to a peak between days 7 and 14. Consistent with reports from the literature [69],[70],[71],[72], these data show a long-lasting release of IL-2 and IL-12 in serum, which could be associated with the reported effectiveness of the treatment. In particular, we observed a decrease in IL-2 levels after day 7, due to the low protein half-life in the serum[73]. Additionally, it is well characterized that helper T cell IL-2 production is limited by negative feedback influenced also by other common gamma chain family cytokines, i.e. IL-4, IL-7, IL-6, IL-12, and IL-27[74]. We cannot rule out that the systemic expression of IL-12 may contributed to lower the basal levels of IL-2 in serum by a negative feedback loop. A sustained high IL-12 concentration in serum could be attributed to functional cooperation between the two interleukins, which leads to a positive feedback loop over the IL-12 expression, as observed by Komel et al.[37].”

Regarding IL-12 serum levels after ECT alone, we have not found information in the literature. However, its behavior in this work is concurrent with the literature regarding IL-12 expression by means of GET.  

Reviewer 3 Report

This study is well-designed and provides novel findings in treating oral melanoma. I would suggest minor revisions before publication.

1. The sample size is small, which should be pointed out as a limitation in the Discussion section.

2. To increase aesthetics, please remove the grid lines in Figures 3 and 4.

Author Response

Responses to the reviewer in bold.

This study is well-designed and provides novel findings in treating oral melanoma. I would suggest minor revisions before publication.

We would like to thank the reviewer for the correction of the manuscript and for the comments provided to improve it.

  1. The sample size is small, which should be pointed out as a limitation in the Discussion section.

We concur with the referee and added the limitations of the study at the end of the discussion section. 

  1. To increase aesthetics, please remove the grid lines in Figures 3 and 4.

We reformatted the Figures to remove the grid lines.

Round 2

Reviewer 1 Report

The authors followed the recommendations and improved the manuscript. It can be accepted.

Reviewer 2 Report

After revision I recommend publishing.